# WNT Signaling in Disease

**DOI:** 10.3390/cells8080826

**Published:** 2019-08-03

**Authors:** Li Fang Ng, Prameet Kaur, Nawat Bunnag, Jahnavi Suresh, Isabelle Chiao Han Sung, Qian Hui Tan, Jan Gruber, Nicholas S. Tolwinski

**Affiliations:** Division of Science, Yale-NUS College, Yale-NUS College Research Labs @ E6, E6, 5 Engineering Drive 1, #04-02, Singapore 117608, Singapore

**Keywords:** WNT, cancer, metabolic syndrome, Alzheimer’s disease

## Abstract

Developmental signaling pathways control a vast array of biological processes during embryogenesis and in adult life. The WNT pathway was discovered simultaneously in cancer and development. Recent advances have expanded the role of WNT to a wide range of pathologies in humans. Here, we discuss the WNT pathway and its role in human disease and some of the advances in WNT-related treatments.

## 1. The Basics of WNT

The WNT signaling pathway is an evolutionarily conserved signal transduction pathway that regulates a wide range of cellular functions during development and adulthood. It controls multiple aspects of development, including cell proliferation, cell fate determination, apoptosis, cell migration and cell polarity during development and stem cell maintenance in adults [1,2,3]. Inappropriate activation of the WNT pathway is also a major factor in human oncogenesis [4].

The first WNT gene, then known as *int-1 (mouse mammary tumor virus integration site 1)*, was isolated from mouse mammary tumors in 1982 [5]. *Int-1* turned out to be highly conserved across multiple species and especially similar to the *Drosophila* gene *wingless* (Wg) a gene found to be involved in wing development, segmentation and formation of body axis during flight development [6,7,8,9]. The name *WNT* comes from a fusion of *wg* and *int* [10,11]. Most animals have several WNT genes with mice and humans encoding 19 WNT genes, seven in *Drosophila* and five in *Caenorhabditis elegans* (*C.elegans*) [6].

The WNT proteins are secreted, lipid-modified glycoproteins, usually 350–400 amino acids in length [12]. WNT proteins act as ligands interacting with Frizzled (FZD) receptors on the cell surface to activate intracellular signaling pathways [6,12,13]. Frizzled receptors are seven-pass transmembrane proteins (similar to G-protein coupled receptors or GPCRs) that act as primary receptors for WNT signals. In addition to the interaction between WNT ligands and FZD receptors, a variety of co-receptors, such as, the low-density lipoprotein receptor related protein (LRP) may be required to mediate WNT signaling [14,15]. Upon activation, a signal is transduced by the pathway activating protein Disheveled (Dsh or Dvl) [13,16]. Dsh proteins have highly conserved protein domains comprised of an amino-terminal DIX domain (named for Dsh and Axin), a central PDZ domain (named for postsynaptic density-95, discs-large and zonula occludens-1) and a carboxy-terminal DEP domain (named for Dsh, Egl-10 and pleckstrin). Dsh acts as a key switch in WNT signaling where, depending on which of the three domains is activated, the WNT signal can be branched off into multiple downstream pathways [16]. The resulting pathways can be categorized into the canonical WNT pathway (β-catenin dependent pathway) and the non-canonical WNT pathways (β-catenin independent pathways) which include polarity and the WNT/Ca^2+^ pathway [15].

**Canonical pathway:** The stability of cytoplasmic β-catenin is mediated by a multimeric protein complex known as the destruction complex formed by the scaffolding proteins (Axin) [17], the tumor suppressor adenomatous polyposis coli (APC) [18,19], glycogen synthase kinase 3 (GSK-3) and casein kinase 1 (CK1). In the absence of canonical WNT ligands, β-catenin binds to the destruction complex and is phosphorylated by CK1 and GSK-3. Phosphorylated β-catenin is then ubiquitinated and subsequently degraded through the proteasome (Figure 1) [14,20,21,22].

In the presence of canonical WNT ligands, β-catenin phosphorylation and degradation is inhibited. Gene expression downstream of canonical WNT signaling is regulated by controlling the amount of the transcriptional co-activator β-catenin. When WNT ligands bind to FZD receptors and LRP co-receptors, a cascade of events is initiated, resulting in the disassembly of the destruction complex stabilizing β-catenin [22]. The signaling cascade is initiated when Dsh is activated through the DIX and PDZ domain, and the FZD-associated Disheveled (Dsh) protein then binds to Axin; Axin then inhibits GSK-3 phosphorylation of β-catenin in the destruction complex. Axin and the destruction complex are thus recruited to the plasma membrane forming signalosomes [23]. β-catenin accumulates in the cytoplasm and localizes to the nucleus where it forms a nuclear complex with DNA-bound T-cell factor/lymphoid enhancing factor (TCF/LEF) transcription factor, resulting in the activation of WNT-responsive genes [14,16,20,24].

**Non-canonical pathways:** In contrast to the β-catenin-dependent canonical pathway, WNT proteins are able to activate additional signaling pathways that are independent of β-catenin. These pathways are called non-canonical pathways, which can be further categorized into two distinct branches, the planar cell polarity (PCP) pathway and the WNT/Ca^2+^ pathway (Figure 2).

The non-canonical PCP pathway regulates a variety of cellular behaviors including planar cell polarity, cell movements during gastrulation and cell migration of neural crest cells [25,26,27,28]. This PCP pathway is activated through the binding of WNT ligands to the FZD receptor independently of the LRP [16]. The non-canonical pathways do not need to utilize the majority of canonical pathway components including WNT itself, but most do involve Dsh and specifically the PDZ and DEP domains and the Disheveled-associated activator of morphogenesis 1 (DAAM 1). This links FZD and Dsh to the small GTPases Rho, and Rho further activates Rho-associated kinase (ROCK), thus leading to cytoskeletal reorganization [29]. Dsh utilizes the DEP domain to form a complex with Rac GTPase, independent of the DAAM 1, then stimulates Jun kinase (JNK) activity and mediates profilin binding to actin [16,28,29]. The polarity signaling pathway is likely best described as a collection of WNT-dependent and WNT-independent effects on cellular organization both within a cell sheet and within the cell itself, showing an intricate interplay between polarity and adhesion [30,31,32,33,34,35].

A second branch of non-canonical WNT signaling is termed the WNT/Ca^2+^ pathway. The role of the WNT/Ca^2+^ pathway is to regulate the release of the intracellular Ca^2+^ from the endoplasmic reticulum [36,37] and modulate signaling for dorsal axis formation and PCP signaling for gastrulation cell movements, cell adhesion, migration and tissue separation during gastrulation [15,38]. Similar to other WNT pathways, the activation of the WNT/Ca^2+^ pathway requires the binding of the WNT ligands to the FZD receptor. The activated FZD receptor directly interacts and activates the heterotrimeric G-proteins, leading to an increase in the intracellular Ca^2+^ concentration [15]. The released Ca^2+^ then activates calcium/calmodulin-dependent kinase II (CamKII), calcineurin or Protein Kinase C [36].

## 2. The Importance of WNT Signaling

Since the identification of the WNT gene more than 30 years ago, various avenues of research have shown roles in myriad processes. The WNT pathway is most famously known for its involvement in hereditary familial adenomatous polyposis (FAP), where a mutated APC tumor suppressor gene fails to regulate β-catenin regulation, allowing tumor cells to progress towards malignancy [39,40]. Among many functions during development, mutations in the WNT pathway disrupt segment polarity in *Drosophila* embryos [41], regulate cardiac development in mice [42] and many other developmental processes in vertebrates and invertebrates [43].

Beyond embryonic development and cancer progression, the WNT pathway has emerged as a key contributor to Alzheimer’s and metabolic diseases. For example, amyloid-β (Aβ) neurotoxicity in Alzheimer’s disease results in downregulated WNT signaling [44], which suggests that downregulated WNT may play an important role in the pathogenesis of Alzheimer’s. WNT signaling has also been linked to metabolic disorders (reviewed by Sethi et al. [45]), where dysregulated WNT is hypothesized to be responsible for obesity and insulin resistance.

Given the vital role that WNT signaling plays in a variety of diseases, this review aims to summarize recent advances in WNT signaling to provide a thorough understanding of WNT signaling in disease.

## 3. WNT Signaling in Aging

Aging can be defined as the time-related progressive accumulation of detrimental changes that are associated with increasing susceptibility to disease and death [46]. In general, aging is a complex biological process associated with a decline in efficiency of physiological processes which include biological function on the molecular, cellular and tissue level [47]. These physiological processes reduce the efficiency of body metabolism, resulting in the disruption of body functional processes and ultimately death. There are a variety of different mechanisms that are thought to participate in the aging process. The WNT signaling pathway is one pathway that may contribute to aging.

Senescence contributes to tissue aging [48] through inhibition of cell differentiation, apoptosis and cell division [49,50], and one of the functions of WNT signaling is to maintain proliferation of tissue stem cells. Ye et al. reported that the canonical WNT2 ligand and downstream canonical WNT-signals are repressed in senescent human cells [49], where downregulation of WNT signaling activates senescence-associated heterochromatin focus (SAHF) assembly, a specialized domain of facultative heterochromatin that represses expression of proliferation-promoting genes, thereby contributing to senescence-associated cell cycle arrest [51]. The formation of SAHF is dependent on GSK3β activity, where increased GSK3β activity decreases the level of β-catenin in senescent cells, suggesting that the canonical WNT pathway is repressed in senescent human fibroblasts [49].

Another example of the detrimental effect of WNT signaling on aging comes from a mouse model of accelerated aging. The authors utilized *klotho* knockout mice that exhibit many age-related disorders [52]. *Klotho* is a transmembrane protein that acts as an anti-aging hormone [53]. *Klotho* binds to WNT proteins and reduces β-catenin levels leading to lower levels of WNT signaling which contributes to stem cell depletion and aging [52,54].

Hair graying is another hallmark of the aging process. One mouse study found that β-catenin expression is significantly elevated in the skin of aged mice through WNT10b/β-catenin signaling promoting melanocyte stem cell differentiation. The authors concluded that the increase in WNT signaling is insufficient to induce hair regeneration but may promote melanocyte stem cell differentiation resulting in a decreased number of melanocyte stem cells and eventually hair graying [55].

In contrast, a number of studies have also suggested that WNT signaling has positive effects on aging. In one study, Chen et al. reported that β-catenin plays an important role in the early phase of fracture healing. Several WNTs (e.g., WNT4, WNT10b and LRP6) were expressed during fracture repair showing activation of WNT/β-catenin signaling through a TCF-dependent transcription reporter in both bone and cartilage formation during fracture repair [56]. Another study found that activation of WNT signaling is required to regenerate hair follicles in wounded mice [57] raising the possibility of therapy using modulators of the WNT pathway to improve bone healing or treating hair loss. As with most WNT studies, these results show a complex role for WNT as it has both positive and negative consequences for aging [58].

## 4. WNT Signaling in Cancer

The WNT pathway plays a complex role in cancer development. Often, mutations of key components are associated with processes like uncontrollable cell proliferation, epithelial–mesenchymal transition (EMT) and metastasis. This section will discuss the main WNT components that have been investigated in oncology and will review the current progress of their roles in cancer development (Figure 3).

**Frequent APC mutations in colorectal cancer:** The APC protein is an important component of the destruction complex in canonical WNT signaling. In addition to this role, APC is also essential for the rapid transition of Axin immediately after WNT stimulation and facilitates the association of Axin and co-receptor LRP6/Arrow in *Drosophila* [59].

As APC negatively regulates the canonical WNT pathway, it is likely that it can act as a tumor suppressor. Consistent with this idea, APC mutations, which are present in approximately ~80% of colorectal cancers, are indeed mostly loss of function and/or truncating mutations [60,61]. These APC mutations frequently occur in the mutation cluster region (MCR) between codons 1285 and 1513 which accounts for only 10% of the entire coding region [62]. To illustrate this, a study concluded that 28 out of 43 (65%) somatic mutations in colorectal cancer cells occur in the MCR [62]. These mutations can inhibit β-catenin ubiquitination and cause uncontrollable transcription [63]. Interestingly, an analysis of 630 human sporadic colorectal cancer tumors revealed that different APC mutations can result in different levels of canonical WNT signaling, and each region of the large intestine has its own optimal threshold of WNT signaling for tumorigenesis [60]. APC mutations also occur in the germline and are present in up to 85% of patients with classical familial adenomatous polyposis (CFAP). Most germline mutations are also truncating (e.g., frameshift, nonsense) [64], confirming APC’s tumor suppressor role.

The prevalence of *APC* mutations in colorectal cancers suggest *APC* as a powerful target for therapy. TASIN-1 (truncated APC selective inhibitor-1) was identified as a molecule that could specifically kill cells with truncated APC’s. TASIN-1′s administration into xenograft and mouse models has shown its effectiveness in tumor suppression and its specificity in killing cells with truncated APC’s while sparing normal cells in vivo [65]. Considering the high frequency of mutated *APC* in patients with colorectal cancer, further studies are needed to investigate APC’s molecular mechanism in β-catenin degradation, its use as a potential biomarker and other possible therapeutic targeting approaches.

**AXIN1/2:** Axin proteins (AXIN1 and AXIN2) are important scaffold proteins that help assemble the β-catenin destruction complex [66]. AXIN1/2 act as negative regulators of β-catenin levels and tumor suppressor proteins. This is consistent with the finding that the overexpression of AXIN*1* inhibits cell growth in hepatocellular carcinoma (HCC) [67]. However, the role of AXIN1/2 as a tumor suppressor through the canonical WNT pathway remains controversial, as other findings have suggested that most human *Axin1* mutated HCCs do not show a β-catenin activation program. These HCCs might have occurred independently of the WNT/β-catenin pathway and instead involved other pathways like YAP and Notch [68].

*A*XIN*2* has been shown to behave as both a tumor suppressor and an oncogene. Its mutations are also observed in various cancers. Chapman et al. observed genetic changes (mostly deletions) of *A*XIN*2* in 7% of human adrenocortical adenomas tumor samples and 17% of adrenocortical carcinomas tumor samples [69]. Consistent with the idea that *A*XIN*2* might act as a tumor suppressor, *A*XIN*2* downregulation is associated with poorer overall survival of patients with breast cancer [70]. However, in vivo findings also suggest *A*XIN*2* as tumor-promoting, as it upregulates the transcriptional repressor, SNAI1, leading to increased EMT and metastatic activity in colorectal cancers in mice [71]. As these contradictory studies provide only preliminary evidence of the functional significance of Axin in canonical WNT pathways, future studies aimed towards exploring Axin’s expression in different cancers and its mechanism in affecting tumor progression should be conducted. The contradictory findings in cancer are not surprising as Axin’s role in development is complex [72].

**LRP5/6:** LRP5 and LRP6 in the low-density lipoprotein receptor (LDLR) family are single-pass transmembrane coreceptors in the WNT canonical signaling pathway. The FZD receptor may form a WNT-induced FZD–LRP6 (or LRP5) complex with these coreceptors. After WNT ligands bind to these receptors, the β-catenin signaling is initiated [73,74].

LRP5 has contradictory roles in cancer progression as well. When osteosarcoma cells are transfected with dominant negative, soluble LRP5 (sLRP5), epithelial–mesenchymal transition (EMT) is reversed, suggesting wildtype LRP5′s role in promoting EMT and metastasis [75]. However, Horne et al. observed an opposite effect in which dominant-negative *LRP5* failed to block osteosarcoma cell formation, suggesting LRP’s role as a tumor suppressor [76].

Similarly, LRP6′s role in cancer progression has been controversial. A total of 45% of human hepatocellular carcinoma cells have overexpressed *LRP6* and as a result, increased β-catenin levels, suggesting LRP6 as tumor-promoting [77]. Consistent with this finding, overexpression of *LRP6* is also found in triple negative breast cancers. When LRP6 is knocked-down in triple negative breast cancer cells, tumor growth is suppressed in vivo [78]. LRP’s also seem to have a promoting role in breast cancer metastasis to bone. TM40D-MB breast cells which are highly metastatic to bone have higher mRNA levels of LRP5, LRP6 and β-catenin, as compared to TM40D cells, which are breast cancer cells that are non-bone metastatic [79]. However, contrary to previous studies, Ren et al. reported the novel role of *LRP5/6* as suppressing metastasis. This study showed that *LRP5/6* downregulation is crucial for metastasis in mouse breast cancer models [80], suggesting that the binding of LRP5/6 to FZD inhibits the FZD-regulated non-canonical pathway and its further tumor metastasis.

Therapeutic targeting of LRP5/6 shows promising results. Mesd, a specialized chaperone that binds to LRP5/6 on the cell surface, inhibits LRP5/6 ligands and suppresses downstream WNT/β-catenin signaling in prostate cancer [81]. Niclosamide, another inhibitor targeting LRP6 on the cell surface, also induces cancer cell apoptosis [81]. Future studies can aim to discover new therapeutic approaches targeting LRP5/6 and other members in the LDLR family. Other LDLR family members like LRP8 and LRP10 are also good potential targets to be further investigated given that previous studies have showed their involvement in breast cancers and hepatocellular carcinomas, respectively [82,83].

**WNT 5A:** The WNT5A ligand binds to certain receptors (e.g., ROR2, ROR1, etc.) and activates non-canonical WNT signaling pathways [84]. Mutations of WNT5A have been shown to associate with cancer development, possibly through noncanonical WNT signaling pathways [84]. For example, WNT5A promotes cancer through binding its receptor ROR2 and enhancing human osteosarcoma invasiveness [85]. *WNT5A* expression induces EMT in a PKC-dependent pathway in human melanoma cell lines [86], suggesting that the increased ability of cells to undergo EMT promotes their invasiveness. Kanzawa et al. reported that the expression of *WNT5A* in human gastric carcinoma-derived MKN-7 cells promoted cancer cell invasiveness by upregulating a transcription factor involved in EMT, namely SNAI1. This study also suggested that *WNT5A* may play a role in creating favorable tumor microenvironments and inducing cancer stem cell properties [87]. Aside from its role of promoting cell invasion, *WNT5A* drives pseudo-senescence in melanoma cancer cells. When exposed to stress, pseudo-senescent cells display classical senescence markers, but are highly invasive, metastatic and resistant to therapy. In response to stressors, tumor cells with knocked down *WNT5A* do not display this pseudo-senescent phenotype, whereas highly expressed *WNT5A* tumor cells do, suggesting WNT5A’s role in promoting cancer cell adaptiveness under stress [88]. WNT5A has also been shown to be involved with cancer cell metabolism and inflammation [89]. Future studies should investigate other WNT ligands (e.g., WNT11) and their roles in cancer progression through non-canonical pathways with the hope of providing novel targets for therapies.

**RNF43:** Ring finger protein 43 (RNF43) is an E3 ubiquitin-protein ligase that degrades WNT receptors such as FZD and LRP5/6. It serves as a negative regulator of the WNT pathway. The mechanisms of WNT signaling suppression are different between non-canonical and canonical pathways. In non-canonical pathways, the suppression involves the interaction between RNF43′s C-terminal cytoplasmic region and Dsh’s PDZ domain. Suppression of the canonical pathway involves FZD’s extracellular protease-associated domain, cysteine-rich domain and the intracellular ring finger domain of RNF43 [90]. A recent study has suggested a novel way that RNF43 suppresses the pathway in the nucleus where a physical interaction between RNF43 and TCF4 translocates TCF4 to the nuclear membrane, inhibiting TCF4 transcriptional activity [91].

*RNF43* acts as a tumor suppressor. When 185 human colorectal tumor samples were analyzed, there were *RNF43* somatic mutations in over 18% of colorectal adenocarcinomas and endometrial carcinomas. This study also showed that most *RNF43* mutations (73%–75% of non-silent mutations) found were truncating, confirming that *RNF43* acts as a tumor suppressor [92]. *RNF43* mutations were also found in pancreatic cancers [93] and mucinous ovarian carcinomas [94]. However, few studies have been conducted to explore the role of *RNF43* in these cancers, so further studies are needed to investigate how certain mutations in *RNF43* can possibly lead to the development of these cancers.

**TCF4/TCF7L2:** T-cell factor/lymphoid enhancer-binding factor (TCF/LEF) transcription factors are the downstream effectors of the WNT pathway. When WNT ligands bind to upstream receptors, β-catenin is released from the cytoplasmic destruction complex and moves to the nucleus, where it associates with TCFs to regulate the transcription of target genes [95].

TCF4, one of the most studied members of the TCL/LEF family, seems to have contradictory roles as both tumor-promoting and tumor-suppressing. TCF4 was originally believed to be tumor-promoting. Van De Wetering et al. demonstrated that dominant-negative TCF4, which fails to bind to β-catenin, halted cell proliferation in colorectal cells [96]. However, Angus-Hill et al. reported TCF4 as tumor-suppressing as its loss of function increased proliferation in colon tumors in mice [97]. This controversy may be resolved by considering that TCL/LEF undergoes alternative splicing, and the isoforms produced may differ in functional domain composition. Thus, these TCL/LEF isoforms will differ in DNA binding and target gene activation [98]. To illustrate this, Tsedensodum et al. identified 14 TCF4 isoforms from four hepatocellular carcinoma cell lines. Functional analysis showed that one isoform, TCF-4K is tumor growth promotive, and another isoform, TCF-4J, is tumor growth suppressive. Strikingly, these two isoforms only differ by five amino acids [99]. These data suggest that the variants caused by alternative splicing may be a reason to explain the contradictory role of TCF4 [100]. Further studies should elucidate the roles of TCF in cancer and development [101], taking into consideration not only the downstream target genes’ roles, but also the different isoforms that result from alternative splicing. Based on this understanding, more effective therapeutic approaches can be developed to aim for certain isoforms that cause malignant phenotypes.

**PTK7:** Another gene linking the various types of WNT signaling to cancer is the gene *PTK7*, an orphan receptor linked to a variety of cancers as a highly upregulated biomarker [102]. PTK7 was originally discovered as a colon carcinoma kinase-4 upregulated in cancer tissue [103]. Its function remained unclear until the ortholog was discovered as a neuronal pathfinding gene in *Drosophila* (known as off-track in flies) [104] and subsequently shown to be a planar polarity gene during mouse development [105]. The mechanism of action remains controversial with WNT and non-canonical functions proposed [105,106,107,108], but the use of PTK7 in cancer treatment could well be under way [109].

**WNT pathway as a tumor suppressor:** Although WNT is usually referred to as an oncogenic pathway, it can also function in quite the opposite direction depending on cellular context. This has primarily been studied in malignant melanoma, where activation of WNT reduces cell proliferation and can actually function as a tumor suppressor. Overall, the main lesson from these studies is that context is crucial, namely depending on what transforming factor is driving oncogenicity since WNT activation can lead to unexpected consequences [110,111,112].

## 5. WNT Signaling in Alzheimer’s Disease

Alzheimer’s disease (AD) is an age-associated neurodegenerative disease (ND) characterized by progressive loss of cognitive function, memory and other associated neurobehavioral issues [113,114]. Extracellular amyloid deposits and intracellular neurofibrillary tangles are the pathological hallmarks of the disease [113]. Mutations in amyloid precursor protein (APP), presenilin-1 (*PSEN1*) and presenilin-2 (*PSEN2*) genes are known to play causative roles in the disease pathogenesis of Alzheimer’s [115,116,117,118]. Alois Alzheimer first described this neurodegenerative disease and characterized the pathological features of AD [115]. Recent genome-wide association studies have identified several risk loci pointing to the involvement of signaling pathways leading to investigations of the influence of specific pathways on the onset and progression of the disease [119,120,121].

The WNT signaling pathway is fundamental to the development of the central nervous system and is strongly linked to AD pathogenesis [122]. WNT signaling plays important roles in the adult brain and regulates synaptic plasticity and memory processes [123]. Impaired WNT signaling has been reported by many animal models and AD clinical studies. For example, a single nucleotide polymorphism in LRP6 is associated with late onset AD making LRP6 a susceptibility gene [124]. Another recent study reported a loss of WNT activity in AD patients showing decreased β-catenin levels and increased phosphorylation of GSK3β in the cortical lobes of AD brains compared to age-matched controls [125]. The WNT inhibitory factor Dickkopf-related protein 1 (DKK1) has been linked to AD as increased DDK1 activity leads to a reduction in WNT signaling and a subsequent decline in cognition has been reported in AD brains [126,127].

Tapia-Rojas and colleagues showed that WNT signaling regulates APP processing by changing the expression of β-secretase (BACE1) enzyme [128]. Inhibition of BACE1 reduces Aβ levels in plasma and cerebrospinal fluid [129]. Overall, these findings indicate that inhibition of WNT/β-catenin signaling promotes amyloidogenic APP processing, Aβ1-42 formation and aggregation. The important pathological hallmark, amyloid deposition, is caused by a compromised blood–brain barrier and an imbalance between deposition and clearance of Aβ. Microglia play a crucial role in the housekeeping of brain regions and also in Aβ clearance by phagocytic and digestive mechanisms. Regulation of phagocytosis and survival of microglia during the process of Aβ clearance is regulated by an innate immune receptor, TREM2, which activates WNT/β-cat signaling, suggesting the involvement of WNT in Aβ clearance [130]. In-depth studies are needed to confirm the involvement of these underlying WNT mechanisms in AD pathology.

WNT signaling has also been linked to tau microfibrillar phosphorylation, specifically through the finding that mutations in *PSEN1* promote GSK3 activity and tau phosphorylation [131], and GSK3-mediated tau phosphorylation upon WNT inhibition [132,133].

An important clinical consequence of AD is cognitive decline and memory. WNT signaling plays a role in the processes of learning and memory through regulation of synaptic structure and function [134,135]. Activation of WNT can result in cognitive improvement as seen in hippocampus-dependent cognitive impairment recovery upon activation of WNT by lithium and rosiglitazone in mouse models [136,137]. These findings suggest that loss of canonical WNT signaling is a key factor in the cognitive aspects of Alzheimer’s disease [138,139].

## 6. WNT Signaling in Metabolic Diseases

WNT signaling is an important regulator of development and progression of organ growth and cell fate. Genes encoding WNT ligands are expressed in the pancreas. Many in vivo and in vitro studies have shown that components of the WNT pathway are involved in pancreatic β-cell proliferation, lipid metabolism and glucose-induced insulin secretion [140,141,142,143]. Type 2 diabetes (T2D) is an important metabolic disorder characterized by persistent hyperglycemia and characteristic insulin resistance. Along with hyperglycemia and insulin resistance, this complex polygenic disease is a collection of metabolic conditions such as glycosuria, hyperlipidemia, neuropathy, nephropathy, retinopathy and negative nitrogen balance. All these metabolic conditions result in defects in insulin secretion and/or insulin action [144,145,146]. Other forms of diabetes include type 1 (T1D) and monogenic forms. T1D is insulin-dependent and treated with insulin while T2D results from insulin resistance and is treated with insulin sensitizing agents. Monogenic forms of diabetes are caused by single gene mutations and the treatment can be tablets and/or insulin.

Recent genome-wide association studies have suggested several metabolic connections and the possible underlying signaling pathways in the pathophysiology of several metabolic diseases and neurodegenerative diseases. The major finding of these human genetic studies is identification of *TCF7L2/TCF4* as a susceptibility gene for T2D. Polymorphisms of *TCF7L2* show a significant effect on the risk of developing T2D [147,148,149]. Natural variants of LRP5 have been shown to be associated with obesity while the missense mutations of LRP6 are associated with the risk of bone loss, early coronary disease and metabolic syndrome [150,151]. Single nucleotide variants of WNT5B are associated with T2D [152]. LRP5 encodes a co-receptor of WNT ligands and the polymorphisms are associated with T1D [153,154] and obesity [155]. These observations show that WNT signaling is involved not only in pancreatic islet development during embryogenesis, but also in the function of the pancreas and intestinal endocrine cells during adulthood. A possible mechanism may be through the accumulation of Reactive Oxygen Species and activation of JNK signaling pathway leading to an increase in nuclear FOXOs [156]. FOXOs and TCF7L2 compete for the limited pool of β-catenin leading to reduced WNT activity which is essential for lipid and glucose metabolism, pancreatic β-cell proliferation and function and the production of GLP-1 (incretin hormone) [157].

Studies have shown the involvement of the WNT pathway in diabetic neuropathy, one of the long-term complications of diabetes [158]. WNT signaling affects numerous cell types like embryonic stem cells, neural cells and mammary cells [159]. Reduction of WNT-1 levels has been reported in diabetic patients [158]. Another link is GSK3 [160] which is involved in mediating inflammation in peripheral and central nervous systems [161,162,163]. One important finding is the link between GSK3 and an increase in insulin receptor phosphorylation in diabetic neuropathy subjects [164] and diabetic neuropathy [165,166,167]. This makes GSK3 a promising target for treatment in many complex diseases such as peripheral diabetic neuropathy [168], neural protection [166] and neuropathic pain reduction [163]. Another link is the association between hypertension, insulin signaling pathway and canonical WNT signaling, where downregulation of canonical WNT signaling in the nucleus tractus solitarii correlates with an increase in phosphorylation of insulin signaling components [169]. These findings bridge the iterations in complex cardiovascular neural pathways.

A long-term complication of diabetes is nephropathy, a process linked to WNT/β-catenin signaling [170]. The hallmark of diabetic nephropathy includes excessive deposition of extracellular matrix proteins in the mesangium, tubulointerstitium of glomerulus and basement membrane, leading to mesangial expansion and renal fibrosis [171]. WNT/β-catenin signaling plays a critical role in the development of diabetic neuropathy [172], but downregulation of WNT/β-catenin signaling leads to adverse effects on the kidneys including increased apoptosis of mesangial cells, increased deposition of fibrous tissue in mesangium, EMT and podocyte injury, renal injury and fibrosis [172,173,174,175,176]. Together these studies point to the importance of WNT signaling in the pathophysiology of diabetes and support the role of WNT signaling as a potential therapeutic target.

## 7. WNT Signaling in Other Diseases

**WNT signaling in cardiovascular diseases:** WNT signaling is essential for heart development, but dysregulation of WNT can cause cardiac and vascular diseases [177,178]. For example, WNT signaling is a major contributor to the development and progression of atherosclerosis, the main cause of cardiovascular disease. A mutation in the LRP6 gene is responsible for autosomal dominant early coronary disease, hypertension, hyperlipidemia and osteoporosis [150], and is a risk factor for carotid artery stenosis in hypertensive patients [179]. The LRP5/6 inhibitor, DKK1, is increased in the plasma and lesions of patients with coronary artery disease and patients with carotid plaques [180]. DKK1 is highly expressed in platelets inhibiting WNT/β-catenin signaling in endothelial cells and leads to endothelial dysfunction [180]. The WNT5A protein, a protein associated with inflammatory disorders [181], was also found to be expressed highly in the macrophage-rich regions of human and murine atherosclerotic lesions [182]. Moreover, WNT5A mRNA and protein levels correlated with advanced arterial lesions [181,183].

Destabilization of an atherosclerotic plaque over time results in the release of plaque’s contents into circulation which can lead to myocardial infarction. Alternatively, plaque erosion with distal embolization or coronary vasospasm can also lead to myocardial infarction [184]. Either way, disruption of blood flow to the region of the heart results in deprivation of nutrients and oxygen, causing the death of cells. WNT signaling is negligible in the normal adult heart, however, pathological conditions trigger the reactivation of fetal genes including WNTs [185]. For example, a mouse model of myocardial infarction showed the dysregulation of expression of WNT2, -4, -7b, -10b and -11 as well as FZD1, -2, -5, -8 and -10 [186,187,188,189], and mice mutants for Dvl1 are more susceptible to infarct rupture after induction of myocardial infarction [178]. FZD2 expression was also found to be upregulated in a cardiac hypertrophy model of rats [178]. Overexpression of FrzA, an antagonist of the WNT/Frizzled pathway, reduced infarct size and improved cardiac function in mice, indicating a role for WNT signaling in neovascularization in the infarct area [186]. An anti-inflammatory and antiapoptotic role for sFRP5 has also been described in a mouse model of ischemia/reperfusion damage [190], supporting a beneficial effect of sFRPs on infarct healing. sFRPs are secreted FZD-related proteins containing the cysteine-rich domain of FZD, but not the transmembrane domains. They appear to bind to WNTs in solution working as inhibitors of WNT. sFRP2 was shown to be a paracrine factor secreted from stem cells to activate the expression of anti-apoptotic genes in cardiomyocytes [191].

WNT signaling has a role in conferring pathology in several cardiovascular diseases such as cardiac arrhythmias, cardiac hypertrophy and heart failure. For a detailed review of WNT signaling in cardiac and vascular diseases, please refer to Foulquier et al. [192].

**WNT signaling in neuronal diseases:** Several studies have also reported a causative link between WNT signaling and autism. Mice mutants for Dvl1 and 3 showed reduced β-catenin expression which in turn caused premature deep layer neurogenesis of neural progenitors in specific brain regions during embryonic development. This had an adverse effect on the precise establishment of neural connections in the future prefrontal cortex, which was observed by serious deficits in brain size and social behavior in adults [193]. This deficit was reversed by administration of a GSK-3 inhibitor to activate the canonical WNT pathway in utero.

Fragile-X syndrome (FXS), a heritable form of autism, is caused by a deficiency of fragile-X linked mental retardation protein (FMRP) that results in various degrees of autism-like behavior. FMRP is a negative regulator of protein translation, which was found to reduce the translation of WNT2 mRNA in the brain of *Fmr1* (FMRP gene) knockout mice [194,195,196]. FXS patients also showed reduced expression of WNT7a and consequently β-catenin-dependent signaling [197]. The connection to WNT signaling was recently extended with the discovery of FMRP and β-catenin functioning in WNT-dependent translation regulation [198].

WNT signaling was linked to the pathogenesis of Parkinson’s disease by the dysregulation of expression of WNT pathway components in brain tissue of Parkinson’s disease patients [199]. Further studies showed that activation of the WNT/β-catenin pathway in astrocytes could recover neighboring midbrain dopaminergic neurons post-injury in vivo [200,201]. Hence, inducing WNT/β-catenin signaling in astrocytes could serve as a therapeutic approach to promote recovery of midbrain dopaminergic neurons [202].

Patients with schizophrenia showed altered GSK3 activity [203], as well as increased expression of WNT1, which can lead to synaptic rearrangement and plasticity [204]. Furthermore, several SNPs in FZD3 were reported to be associated with susceptibility to schizophrenia [205].

Surprisingly, a recent study by Paige et al. showed the essential role of the WNT signaling pathway in promoting functional recovery from spinal cord injury in lampreys [206]. These studies signify the importance of WNT signaling in the pathogenesis of neuronal diseases and injury as well as a potential target for therapeutic intervention. The relevance of WNT signaling in depression, bipolar disorder as well as epilepsy and seizures have also been reported [7].

**WNT signaling in liver diseases:** A notable feature of the liver, unlike other tissues, is its ability to regenerate post-injury, driven by the WNT/β-catenin signaling pathway [207,208,209,210]. Loss of β-catenin and other components of the WNT signaling pathway causes delayed liver regeneration following a partial hepatectomy [211,212,213]. Genetic polymorphisms in key members of the WNT-β-catenin pathway (*SFRP2*, *FZD4*, *SOST*, *CTNNB1* and *CSNK1A1*) have also been associated with inflammation and fibrosis in patients with hepatitis C [214]. β-Catenin is upregulated in fibrotic human liver tissue, and increased β-catenin levels in rat hepatic stellate cells were associated with collagen production and proliferation [215]. Conversely, inhibition of canonical WNT signaling pathway by β-catenin/CREB-binding protein (CBP) inhibitor ICG-001 ameliorated fibrosis in vivo through pharmacological inhibition of stromal CXCL12, a potential therapeutic [216].

Mice with mutant LRP6 developed a fatty liver due to increased AKT/mTOR signaling resulting in elevated hepatocyte lipogenesis, which could be reversed by the administration of WNT3a [217]. Moreover, under conditions of oxidative stress, β-catenin bound directly to FOXO enhanced the transcription of FOXO target genes [218]. This interaction also modulated hepatic insulin signaling by promoting the expression of rate-limiting enzymes in hepatic gluconeogenesis, and the liver-specific deletion of β-catenin in mice fed a high-fat diet showed higher glucose tolerance due to decreased gluconeogenesis [219].

Accumulation of toxic cholephilic substances such as bile acids due to reduced bile flow can cause liver damage, a condition known as cholestasis. This is characterized by a reduced bile flow, which often results in the accumulation of toxic cholephilic substances such as bile acids that can cause liver damage [220]. Liver-specific *Ctnnb1*-null mice showed defective bile acid homeostasis with elevated levels of toxic liver compounds in the serum and liver [221,222], suggesting the role of the WNT/β-catenin pathway in regulating bile flow.

Hypoxia inducible factor-1α (HIF1α), which mediated transcription and promotion of cell survival during ischemia/reperfusion injury in hepatocytes, was also found to be mediated via β-catenin signaling [223,224].

**WNT signaling in kidney diseases:** Tubule-specific β-catenin knockout mice showed higher mortality, increased serum creatinine as well as more severe morphologic injury after acute kidney injury as compared to control mice [225]. Conversely, WNT4-mediated activation of β-catenin at different intervals after acute kidney injury results in recovery from renal injury by promoting cell cycle progression [226]. WNT7B derived from macrophages activated WNT signaling to confer protection and trigger the repair process after ischemia reperfusion injury in mice [227]. β-Catenin signaling was also reported to reduce Bax-mediated apoptosis and improve cell survival after induction of metabolic stress in proximal tubular epithelial cells [228]. A WNT agonist was also found to attenuate inflammation and oxidative stress after ischemic reperfusion injury, suggesting the potential pharmacological application of manipulating WNT activity on preventing kidney ischemic-reperfusion injury [229].

Continuous activation of WNT/β-catenin results in the progression of acute kidney injury to chronic kidney disease displaying interstitial myofibroblast activation and excessive extracellular matrix deposition [230]. A mouse model of unilateral ureteral obstruction showed upregulation of the expression of several WNT ligands with consequential upregulation of target genes involved in fibrosis such as fibronectin, plasminogen activator inhibitor-1 and matrix metalloproteinase 7 [172,231,232,233,234,235,236]. Inhibition of β-catenin activation using secreted Frizzled-related protein 4 (sFRP4), an endogenous extracellular WNT antagonist, resulted in reduced renal fibrosis after unilateral ureteral obstruction possibly by preventing WNTs binding to their receptors [237]. Targeted inhibition of WNT/β-catenin signaling using a small molecule peptidomimetic, ICG-001, suppressed matrix expression and ameliorated renal interstitial fibrosis [238,239,240]. A similar upregulation of WNT proteins in kidney specimens from patients with chronic kidney disease was observed, suggesting a role for this pathway in kidney repair and regeneration or progression of the disease in humans as well [241,242]. Hence, ablation of WNT/β-catenin signaling may be an effective approach to ameliorate kidney injury and fibrotic lesions in various models of chronic kidney disease [243].

**WNT signaling in lung diseases:** Increased levels of matrix metalloproteinases (MMPs) and pro-inflammatory cytokines such as IL1, IL6, IL8 and IL15 are general features of lung inflammation. The canonical WNT signaling pathway has been shown to regulate the expression of both MMPs and pro-inflammatory cytokines. Activation of canonical WNT signaling increased expression of MMP-2, MMP-3, MMP-7, MMP-9 and MT3-MMP in mice [244] suggesting that WNT regulates the expression of MMPs, thus regulating lung inflammation.

In addition, the WNT5A-FZD5 signaling cascade may play a role in regulating the levels of pro-inflammatory cytokines. In pathological studies of rheumatoid arthritis, overexpression of WNT5A and FZD5 correlates with increased levels of pro-inflammatory cytokines (IL-6, IL-8, IL-15) [245]. Experimentally, expression of WNT5A in normal fibroblast cells also increased IL-15 mRNA and protein expression in synovial fibroblasts [245], suggesting that WNT5A can activate the expression of pro-inflammatory cytokines. Furthermore, inhibition of WNT5A with antisense WNT-5A vectors and anti-FZD5 antibodies reduced the expression of pro-inflammatory cytokines IL-6 and IL-15 [246]. While the exact pathway for WNT5A-FZD5 to activate the expression of these pro-inflammatory cytokines remains unknown, there is strong evidence to suggest that WNT5A-FZD5 can regulate pro-inflammatory cytokines, which play a key role in lung inflammation. While a comprehensive review of WNT signaling in lung diseases is beyond the scope of this paper, a good summary can be found in [247].

In idiopathic pulmonary fibrosis (IPF), immunohistochemical analyses reveal that MMP7 is consistently upregulated [248]. Higher MMP7 also correlates with a worse prognosis for IPF [249], while mouse knockout models for MMP7 showed decreased pulmonary inflammation [250]. As MMP7 is a target of the canonical WNT signaling pathway [251], it is hypothesized that the canonical WNT signaling pathway will also play a role in the pathogenesis of pulmonary fibrosis.

**WNT signaling in oral disease:** Mutations in WNT signaling have also been associated with oral diseases. In salivary gland tumors, loss-of-function mutations in the WIF1 WNT inhibitor gene are associated with salivary gland oncogenesis [252]. Mutations in the APC gene are also responsible for Gardner syndrome, where numerous jaw and teeth cysts form in addition to intestinal polyposis [253]. Mutations in AXIN2 also result in tooth agenesis (the developmental absence of one or more teeth), with nonsense AXIN2 mutations causing severe permanent tooth agenesis via activating the WNT signaling pathway [254,255].

**WNT signaling in skin disease:** A nonsense mutation in WNT10A is responsible for severe hypodontia in odonto-onycho-dermal dysplasia, a rare ectodermal dysplasia characterized by keratoderma, dry hair and hyperkeratosis of the skin [256]. Mutations in the human *PORCN* (porcupine) gene, a regulator of WNT signaling, result in focal dermal hypoplasia, characterized by widespread lesions of dermal hypoplasia [257]. In psoriasis, WNT5A is upregulated while the WNT Inhibitory Factor (WIF) is downregulated [258], and pathway analysis revealed that this may be due to activation of non-canonical pathways.

**WNT signaling in allergic airways:** Impaired WNT signaling is also associated with diseases of allergic airways. In a mouse model, expression of WNT suppressed the development of allergen airway disease, while treatment with LiCl (a mimic of canonical WNT signaling) protected wild-type mice from development of allergic airway diseases [259]. In asthma, airway smooth muscle (ASM) cells of patients show upregulated WNT-5A [260], suggesting that the WNT pathway could be a potential target for therapeutics.

**WNT signaling in skeletal diseases:** The notion that WNT is crucial to bone development gained traction in the early 2000s, when several studies reported that an inactivating mutation in LRP5 was associated with osteoporosis-pseudoglioma syndrome (OPPG), characterized by lower bone mass and strength and a higher predisposition to suffer from bone fractures [261]. Another group discovered that an inherited G171V mutation in LRP5 led to increased bone density, which was supplemented by mouse models that showed similar increases in bone density and osteoblast numbers [262,263]. These findings highlighted the importance of LRP5 for bone development and pioneered the interest in WNT’s role in bone development and disease.

The molecular mechanism by which WNT modulates bone growth and repair is known. WNT regulates the differentiation of osteoblasts and chondrocytes, thereby directly affecting pathologies such as osteoarthritis, a disease characterized by dysregulation of osteoblast formation and bone remodeling [264]. Sclerostin acts as a WNT antagonist by competitively binding to LRP5, resulting in decreased bone masses [265]. A follow-up study then highlighted that the G171V mutation in LRP5 interfered with sclerostin binding, providing a mechanistic explanation for sclerostin and LRP5 interaction in bone development [266]. Dickkopf-1, a well-known WNT antagonist, similarly had its inhibitive capacities decreased in G171V mutants [267]. The inability for DKK1 and sclerostin to inhibit LRP5 results in increased bone mass and mineral content and provides a very clear foundation of understanding how WNT signaling functions in relation to bone development.

As WNT regulation is important to bone development and repair, it is not surprising that dysregulation of the pathway leads to bone disease. As studies that intersect WNT signaling and bone development continue to emerge, novel interactions of WNT mutations that underlie bone-related pathologies will also surface. So far, studies have shown involvement of the WNT pathway in osteoarthritis and osteoporosis [268,269,270], while more recent studies uncovered that mutations in WNT1 cause osteogenesis imperfecta and early-onset osteoporosis [271,272]. This has resulted in a recent surge of interest in targeting the WNT pathway for therapeutic intervention of skeletal disorders, which we will explore further in the subsequent section of this review.

## 8. Clinical Relevance of WNT Signaling

The previous sections have highlighted the importance of WNT signaling in various pathological and physiological contexts. A general search of “WNT in *x* disease” through academic journals and recent publications will almost always yield results. As such, it has become obligatory to invest resources into targeting this pathway, which unfortunately has had little success.

The general concept of treating diseases through the WNT pathway is through one of two possibilities: either enhancing or inhibiting the pathway. In diseases where WNT is lacking, the inhibition of naturally occurring WNT antagonists or upregulation of WNT agonists can lead to supplementation of WNT signaling levels to yield therapeutic effects [273]. In contrast, in diseases where WNT is aberrant such as metastatic breast and colorectal cancers, the inhibition of the WNT pathway can also lead to therapeutic benefits [274,275].

There are ongoing efforts to find ways of targeting this pathway, mostly for the treatment of various WNT-driven cancers. However, the WNT pathway is a great candidate target for the treatment of pathologies related to musculoskeletal and neurological systems. The WNT pathway has also drawn interest in the field of dermatology. In this section, we present an update on agonists and antagonists of WNT signaling in various clinical contexts.

**Cancers:** Cancers, perhaps, have the highest potential for the development of WNT therapeutics in the clinic. Aberrant activation of WNT through mutations of specific WNT components can be found in colorectal, endometrial, breast and lung cancers, among many others [92,276,277]. Arguably, the classification of cancers as ‘WNT-addicted’ or ‘WNT-driven’ has potential pitfalls with regards to potential treatment options—simply targeting the inhibition of WNT could be considered a naïve approach that culminates in insufficiently addressing the complexity of aberrant WNT signaling in cancers [278,279]. There are ongoing efforts in trying to understand the complexity of WNT in driving tumorigenesis and how it can successfully be targeted. We hereby present an overview of WNT-related cancer drugs that are in development (Figure 4).

### Inhibitors of WNT Pathway Components

Porcupine (PORCN) is a membrane-bound O-acyltransferase involved in the secretion of WNT ligands [280]. There are two inhibitors for PORCN that are undergoing clinical trials, WNT974 and ETC-159. WNT974 is efficacious in blocking WNT signaling and tumor growth, especially in head and neck squamous cell carcinoma cell lines [281]. ETC-159, a small molecule inhibitor of PORCN, is similarly effective in the treatment of preclinical colorectal cancer models with R-spondin mutations [282]. Both drugs are currently undergoing Phase I/II clinical trials for the treatment of colorectal cancers [283].

Another approach currently in clinical trials is the introduction of WNT signaling inhibitors directly into the blood stream of patients. For example, Vantictumab is an antibody targeting 5 of the 10 human FZD receptors (FZD 1, 2, 5, 7, and 8) binding to the extracellular domains of FZD and interfering with signaling [284,285]. Another example is the FZD8-derived WNT scavenger Ipafricept, a fusion protein containing the extracellular ligand binding domain that acts as a decoy receptor [286,287].

Another class of WNT inhibitors that has shown efficacy in the treatment of WNT-related cancers are Tankyrase inhibitors. Tankyrase 1 and 2 belong to the poly ADP-ribose polymerase (PARP) family of proteins and are associated with the WNT pathway by facilitating the degradation of Axin [288,289]. There are currently four Tankyrase inhibitors which are undergoing preclinical studies—XAV939, IWR1, JW74, and NVP-TNKS656 [283]. Interestingly, NVP-TNKS656 treatment of colorectal mouse xenografts and patient-derived cultures re-sensitized the cancers to PI3K and AKT inhibitors [288]. The same research group also showed that WNT hyperactivation was responsible for colorectal cancers acquiring resistance to PI3K or AKT inhibition. Interestingly, the treatment of colorectal xenograft cancers using the Tankyrase inhibitor NVP-TNKS656 led to overcoming WNT-mediated drug resistance. Further studies must investigate the safety of Tankyrase inhibitors for gastrointestinal toxicity [290], but Tankyrase inhibitors show great potential especially in combinatorial therapy or as a second line of drugs to overcome acquired resistance in tumors.

These anti-cancer agents target more peripheral members of the WNT pathway. This is necessary, as the treatment of core WNT components could result in non-selective inhibition of WNT and detrimentally affect WNT-dependent stem cell populations, thereby resulting in unwanted side effects. This is especially true for tissues with high rates of turnover, such as hair follicles and the intestinal crypts, both of which are controlled by WNT signaling. However, there are still several anti-cancer drugs in development which do target components more central to the canonical WNT signaling cascade. For example, the preclinical drug PKF118-310 directly inhibits the β-catenin/TCF transcriptional complex [291]. PRI-724 is another drug that targets β-catenin by inhibiting its interaction with CBP (CREB-binding protein) [292]. When treated in preclinical pancreatic cancer models, PRI-724 promoted differentiation of chemoresistant cancer stem cells and significantly decreased metastatic potential. PRI-724 is currently being evaluated in clinical trials for patients with advanced pancreatic adenocarcinoma [293].

**WNT inhibition via cross-talking pathways:** As with many other signaling pathways, the WNT pathway is highly intricate and complex. This is reflected in its many interacting partners and involvement in various physiological and pathological contexts. An additional feature of this complexity is the crosstalk that occurs between the WNT pathway and other signaling cascades, especially in cancers [294,295,296]. As such, the inhibition of pathways which crosstalk with WNT may lead to therapeutic effects on WNT-driven cancers themselves. We discuss some of the ways in which this idea has been implemented in both preclinical and clinical settings.

sFRP-1 is another potential target for the treatment of WNT-driven cancers. It functions as a mediator of crosstalk between two signaling cascades—Hedgehog and WNT [295]. Vismodegib is an FDA approved drug that binds directly to Smoothed (SMO) and is used in the treatment of advanced basal cell cancers [297]. Erismodegib is a similar FDA-approved SMO antagonist also used in the treatment of basal cell cancers [297].

There are several ongoing efforts to develop therapies to target the aberrant WNT signaling found in many cancers. With drugs ranging from various antibodies of WNT proteins to small molecule inhibitors, the WNT pathway continues to be an attractive yet elusive target for the treatment of cancers due to its overwhelming complexity and the possible indirect consequences to healthy, WNT-dependent cells. We are certain to see more drugs roll out over the course of the next decade as these WNT anti-cancer agents are further studied and some of the clinical trials move towards FDA approval.

**Neurological diseases:** WNT signaling is necessary for the development of the early brain, as it plays crucial roles in determining cell polarity and migration, as well as neural patterning and synapse development [298,299,300]. Mutations that occur to the WNT pathway could lead to brain defects and malformations [299]. The dysregulation of the WNT pathway has also been implicated in various neurological diseases, ranging from schizophrenia, autism, Alzheimer’s and epilepsy [301,302,303]. We discuss some plausible avenues where WNT could potentially be used for treatment of these disorders.

Antagonists of the WNT pathway, such as DKK1 protect against hippocampal sclerosis, a key characteristic of epilepsy of the temporal lobe [304]. As discussed at length in previous sections, a lot of effort was put forth to investigate novel mechanisms to inhibit the WNT pathway. However, relatively few studies explored the effects of these compounds in relation to epileptic seizures [301]. Current treatment options for therapy are diverse, but most do not target the underlying mechanism of epilepsy and only attempt to mitigate seizures. The WNT pathway, which is disrupted following seizures, could therefore be a promising target for epilepsy therapies.

Another neurological disorder where WNT inhibition could potentially be used therapeutically is in cell therapy. More specifically, WNT inhibition could be used to generate cortical motor neurons (CMNs) in order to treat damaged corticospinal tracts, which may occur from various neurodegenerative disorders [305]. The WNT inhibitor WNT-C59, a PORCN inhibitor, induced the differentiation of human iPSCs into cortical neurons [306]. Furthermore, once grafted into the cortex of adult mice, the induced cells showed fiber extension towards the spinal cord and this suggests the potential of using C59 for cell replacement therapy in motor neuron diseases.

**Musculoskeletal diseases:** A strategy that has been proposed to treat musculoskeletal diseases via the WNT pathway is to develop inhibitors for WNT antagonists that are relevant to bone development, mainly DKK1 and sclerostin [269]. Mouse bone fracture models show that DKK1 antibody treatment leads to significant gains in bone mineral content and densities [56,307]. The sclerostin antibody romomosumab (Amgen), which is nearing submission to the FDA for approval, has shown great efficacy in its human trials and functions similarly in increasing bone mineral density [308,309,310].

As the use of WNT therapies for musculoskeletal disorders is a relatively new field, there are many drugs that are presently under preclinical and clinical studies. For instance, SM04690 is a small molecule inhibitor of the WNT pathway and is presently undergoing clinical trials for degenerative disc disease [311]. When tested in vitro, the molecule promoted proliferation, and in its preclinical rodent model, SM04690 had very low toxicity and promoted regeneration of cartilage matrix while improving disc height, health and shape. This molecule is also known as lorecivivint and appears to inactivate the WNT pathway through a novel mechanism involving the nuclear kinases CDC-like kinase 2 (CLK2) and dual-specificity tyrosine phosphorylation-regulated kinase 1A [312,313].

Osteoarthritis is another musculoskeletal disease where the WNT pathway is being explored for potential therapy. Fluoxetine and Verapamil, two pre-approved drugs, were able to successfully suppress abnormal WNT activity and slowed cartilage degradation in vitro as well as in rodent models [270,314]. SM04690, the small molecule inhibitor for degenerative disc disease, was also tested for amelioration of osteoarthritis symptoms. In a double-blind, placebo-controlled phase II study, patients with osteoarthritis who took SM04690 reported significant improvements compared to the control group [311,315]

**Dermatological disorders:** The interest in WNT-related therapies has also made it to the field of dermatology. Androgenetic alopecia (AGA), commonly known as male pattern baldness, is the most common progressive hair loss disorder in men and is characterized by progressive hair follicular miniaturization [316]. SM04554 is a topical scalp treatment that targets the WNT pathway and is presently in Phase II clinical trials. Specifically, the treatment resulted in increased follicle counts and suggests that it may be promoting follicular neogenesis, which is in concurrence with our understanding of the role that WNT plays in hair growth and spacing [317]. Psoriasis is a skin condition that occurs due to excess proliferation of skin cells. SM04755, another small molecule WNT inhibitor, has also shown good efficacy in treating psoriasis in preclinical study rodent models by regulating keratinocyte proliferation [315].

**Complexity of treating diseases via WNT modulation:** WNT signaling is a highly complex pathway regulating many biological processes. WNT-related therapeutics focus on one of two strategies: they either inhibit aberrant WNT signaling or enhance deficient WNT signaling. However, these two strategies are an all-or-nothing approach, which overlooks the complexity of WNT signaling and its effect on different cells. The various WNT ligands and receptors generate permutations that allow for precise control of signaling in contexts ranging from intestinal crypt cell renewal to hair follicle spacing. The intricacy of WNT signaling in various physiological contexts leads to a complexity of disorders and diseases where the various WNTs and receptor-complexes define diseases [273,318].

One approach is to consider the Goldilocks hypothesis (WNT levels being just right) when looking at WNT signaling in diseases. A study found that the level of WNT signaling, the position of the tumor in the colon and the rate of tumorigenesis is highly correlated. For example, APC mutations in the proximal and distal colon vary in their number of β-catenin binding sites required for tumorigenesis [60,319,320]. This suggests that the amount of WNT signaling that is necessary to promote growth varies even along the position of the colon, and that too much intracellular WNT signaling may also be detrimental to cancer growth [321]. The tendency to look for the most potent enhancers or inhibitors may be a flawed approach in treating WNT-related diseases, as we must move forward with a more nuanced understanding of how WNT operates in various physiological and pathological contexts.

## Figures and Tables

**Figure 1 cells-08-00826-f001:**
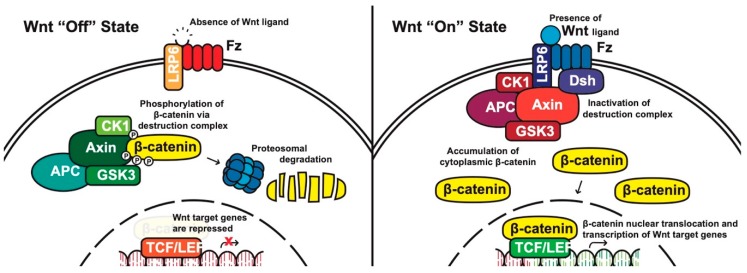
The canonical WNT pathway in on and off states. APC, adenomatous polyposis coli; CK1, casein kinase 1; GSK3, glycogen synthase kinase 3; TCF/LEF, T-cell factor/lymphoid enhancing factor; LRP6, lipoprotein receptor related protein 6; FZD/Fz, Frizzled; Dsh, Disheveled.

**Figure 2 cells-08-00826-f002:**
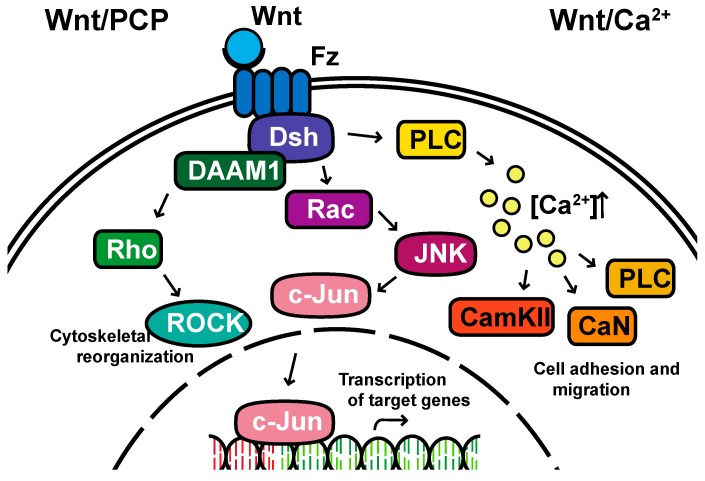
The non-canonical planar cell polarity and Calcium pathways. DAAM1, Disheveled-associated activator of morphogenesis 1; Rac; ROCK, Rho-associated kinase; JNK, Jun kinase; PLC, Phospholipase C; CamKII, calcium/calmodulin-dependent kinase II.

**Figure 3 cells-08-00826-f003:**
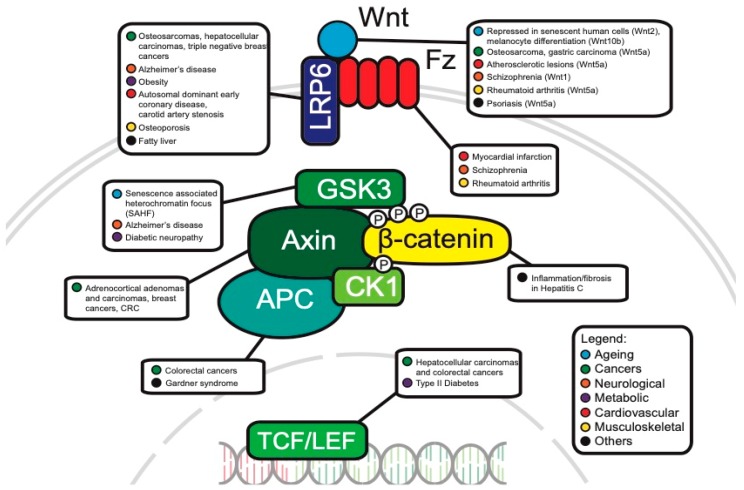
Canonical WNT pathway components associated with disease.

**Figure 4 cells-08-00826-f004:**
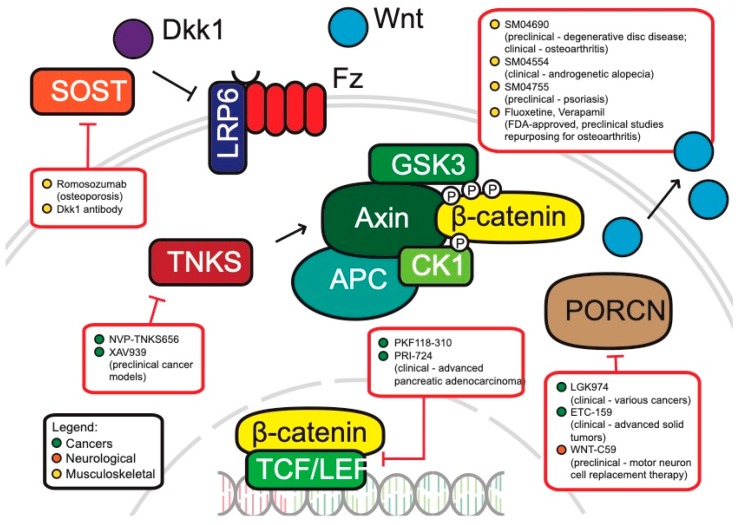
Drugs developed against canonical WNT pathway components. SOST, Sclerostin; PORCN, porcupine; TNKS, Tankyrase.

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
