# Peer review of "WNT Signaling in Disease"

_cells, 2019, doi:10.3390/cells8080826_

Round 1
Reviewer 1 Report
In this review article entitled “Wnt signaling in disease”, Ng and colleagues provide a broad overview of the multiple diseases in which WNT signaling have been implicated. The paper is well-written and the senior author has a clear track record in the field. Due to the broad nature of the review, many subjects are only touched upon briefly. This makes this review a great introduction into WNT-related diseases for the novice in the field. However, I would recommend referring to one or two recent state-of-the-art reviews for every condition you discuss in order to serve as a starting point for readers who wish to gain more in-depth information on a specific topic. Furthermore, I do have some suggestions to improve the manuscript:
1. I suggest using the IUPHAR-approved nomenclature (G. Schulte, Pharm Rev 2010; 62:632): FZD rather than Fz and WNT instead of Wnt. Moreover, there are some inconsistencies in the spelling of WNT (e.g. line 490: WNT5a)
2. Several abbreviations are not specified, e.g. line 291: APP, PSEN1 and -2; line 308: BACE. Please check the entire manuscript for unspecified abbreviations
3. line 269: Van De Watering should read Van De Wetering
4. line 296: [112] [113] [114] should read [112-114]
5. line 337: ‘…T2D is not totally dependent on insulin treatment’. The initial problem in T2D is insulin resistance, not a shortage of insulin. Therefore, T2D is treated with insulin-sensitizing agents. Only in late stages of the disease the addition of insulin may be needed. Could you please revise this statement?
6. line 360: “This makes GSK3 a promising … for treatment of many complex diseases” I think the word ‘target’ is missing here
7. Chapter 8: In this overview of the different ways of targeting beta-catenin mediated WNT signaling, the FZD targeting antibody Vantictumab and the FZD8-derived WNT scavenger Ipafricept could be added as they are in clinical trials for cancer therapy.
8. line 581: the porcupine inhibitor LGK974 is now generally referred to as WNT974
9. Line 581-2: “LGK974 is efficacious in blocking Wnt signaling and tumor growth in vivo, especially in heat and neck squamous cell carcinoma cell lines”. This makes no sense to me, as the cell lines are an in vitro model.
10. line 645: ‘protects’ should read ‘protect’
11. Some references appear to be incomplete regarding the page numbers, e.g. ref #11, #198, #211. Please check.
Author Response
Reviewer 1:
In this review article entitled “Wnt signaling in disease”, Ng and colleagues provide a broad overview of the multiple diseases in which WNT signaling have been implicated. The paper is well-written and the senior author has a clear track record in the field. Due to the broad nature of the review, many subjects are only touched upon briefly. This makes this review a great introduction into WNT-related diseases for the novice in the field. However, I would recommend referring to one or two recent state-of-the-art reviews for every condition you discuss in order to serve as a starting point for readers who wish to gain more in-depth information on a specific topic. Furthermore, I do have some suggestions to improve the manuscript:
The current special issue will provide a variety of in depth reviews on Wnt in specific diseases. Where possible, we have provided reviews for more in depth reading.
I suggest using the IUPHAR-approved nomenclature (G. Schulte, Pharm Rev 2010; 62:632): FZD rather than Fz and WNT instead of Wnt. Moreover, there are some inconsistencies in the spelling of WNT (e.g. line 490: WNT5a)Changed
Several abbreviations are not specified, e.g. line 291: APP, PSEN1 and -2; line 308: BACE. Please check the entire manuscript for unspecified abbreviationsAdded
line 269: Van De Watering should read Van De WeteringFixed
line 296: [112] [113] [114] should read [112-114]Fixed
line 337: ‘…T2D is not totally dependent on insulin treatment’. The initial problem in T2D is insulin resistance, not a shortage of insulin. Therefore, T2D is treated with insulin-sensitizing agents. Only in late stages of the disease the addition of insulin may be needed. Could you please revise this statement?This has been corrected
line 360: “This makes GSK3 a promising … for treatment of many complex diseases” I think the word ‘target’ is missing hereFixed
Chapter 8: In this overview of the different ways of targeting beta-catenin mediated WNT signaling, the FZD targeting antibody Vantictumab and the FZD8-derived WNT scavenger Ipafricept could be added as they are in clinical trials for cancer therapy.This has been added
line 581: the porcupine inhibitor LGK974 is now generally referred to as WNT974Changed to WNT974
Line 581-2: “LGK974 is efficacious in blocking Wnt signaling and tumor growth in vivo, especially in heat and neck squamous cell carcinoma cell lines”. This makes no sense to me, as the cell lines are an in vitro model.In vivo was deleted
line 645: ‘protects’ should read ‘protect’Fixed
Some references appear to be incomplete regarding the page numbers, e.g. ref #11, #198, #211. Please check.Ref 11 is only one page. For 198 and 211 this has been updated.
Reviewer 2 Report
This is an interesting and thorough review on Wnt signaling pathways in disease, and is suitable for publication in Cells after some revision. My only major point is that when talking about Wnt/beta-catenin signaling in cancer, the authors seemed to overly focus on Wnt as an oncogenic pathway throughout the paper except for the last paragraph. In fact, Wnt may function as an oncogenic or tumor suppressor pathway, depending on the context (see a series of reviews by Randall Moon, such as Anastas and Moon, Nat Rev Cancer2013, Zimmerman et al., CSH Perspect Biol2012, and Lucero et al., Curr Oncol Rep2010). This should be discussed in more detail to provide a nonbiased view.
Minor points:
The authors used both “aging” and “ageing”. Either spelling is fine, but they need to be consistent throughout the manuscript.
Lines 117-146 need to be re-organized. Lines 117-126 discuss the positive effects of Wnt on ageing, and belong to the last paragraph (Lines 138-146).
Lines 158-159: It is unclear what “somatic colorectal cancers” means. APC mutations are present in ~80% of colorectal cancers.
Line 263: TCF7L2 is the official name for TCF4, not the whole class of TCF/LEF transcription factors.
Lines 283-286: Roles for PTK7 can be expanded a little more. The authors cited 6 papers but summarized them in just 2 sentences.
Line 329: It is unclear what “Wnt factors” means. The authors need to specify if they were referring to Wnt ligands, Wnt signaling pathway components, or something else.
Line 335: “defects if” should be “defects of”.
Line 337: The abbreviation “T2D” is not defined.
Lines 400-401: A brief description of sFRPs is needed for readers outside of the Wnt field.
Lines 414-418: Clarify the relationship between FMRP (protein name) and Fmr1(gene name; should be italicized). A recent paper linking b-catenin directly to the translational repressor function of FMRP (Ehyai S, et al., EMBO Rep. 2018) should be included.
Line 538: SOSTis the gene that encodes sclerostin, and should be deleted here based on the context.
Line 568: “Enhancers” are DNA sequences that regulate gene transcription, and should be replaced by “activators” or “agonists” here to avoid confusion.
Lines 629-631: It’s unclear if these are related to Wnt signaling. If not, they should be deleted.
Line 675: Explain what SM04690 is.
Author Response
Reviewer 2:
This is an interesting and thorough review on Wnt signaling pathways in disease, and is suitable for publication in Cells after some revision. My only major point is that when talking about Wnt/beta-catenin signaling in cancer, the authors seemed to overly focus on Wnt as an oncogenic pathway throughout the paper except for the last paragraph. In fact, Wnt may function as an oncogenic or tumor suppressor pathway, depending on the context (see a series of reviews by Randall Moon, such as Anastas and Moon, Nat Rev Cancer2013, Zimmerman et al., CSH Perspect Biol2012, and Lucero et al., Curr Oncol Rep2010). This should be discussed in more detail to provide a nonbiased view.
A paragraph on this has been added.
Minor points:
The authors used both “aging” and “ageing”. Either spelling is fine, but they need to be consistent throughout the manuscript.
Fixed
Lines 117-146 need to be re-organized. Lines 117-126 discuss the positive effects of Wnt on ageing, and belong to the last paragraph (Lines 138-146).
We tried to address this, but the issue is that it is positive in one sense but negative in the mutant studied.
Lines 158-159: It is unclear what “somatic colorectal cancers” means. APC mutations are present in ~80% of colorectal cancers.
Corrected
Line 263: TCF7L2 is the official name for TCF4, not the whole class of TCF/LEF transcription factors.
Corrected
Lines 283-286: Roles for PTK7 can be expanded a little more. The authors cited 6 papers but summarized them in just 2 sentences.
This has been expanded to be more historical. The current controversies in the PTK7 field are beyond the scope of a disease review and will hopefully be settled soon.
Line 329: It is unclear what “Wnt factors” means. The authors need to specify if they were referring to Wnt ligands, Wnt signaling pathway components, or something else.
Corrected
Line 335: “defects if” should be “defects of”.
Corrected
Line 337: The abbreviation “T2D” is not defined.
Corrected
Lines 400-401: A brief description of sFRPs is needed for readers outside of the Wnt field.
A brief description has been added.
Lines 414-418: Clarify the relationship between FMRP (protein name) and Fmr1(gene name; should be italicized). A recent paper linking b-catenin directly to the translational repressor function of FMRP (Ehyai S, et al., EMBO Rep. 2018) should be included.
Corrected
Line 538: SOSTis the gene that encodes sclerostin, and should be deleted here based on the context.
Corrected
Line 568: “Enhancers” are DNA sequences that regulate gene transcription, and should be replaced by “activators” or “agonists” here to avoid confusion.
Corrected
Lines 629-631: It’s unclear if these are related to Wnt signaling. If not, they should be deleted.
Deleted
Line 675: Explain what SM04690 is.
A brief description has been added.